# Air pollution impacts from warehousing in the United States uncovered with satellite data

Gaige Hunter Kerr [1] ✉, Michelle Meyer [2], Daniel L. Goldberg [1], Joshua Miller [2] & Susan C. Anenberg [1]

Regulators, environmental advocates, and community groups in the United States (U.S.) are concerned about air pollution associated with the proliferating e-commerce and warehousing industries. Nationwide datasets of warehouse locations, traffic, and satellite observations of the traffic-related pollutant nitrogen dioxide ($NO_2$) provide a unique capability to evaluate the air quality and environmental equity impacts of these geographically-dispersed emission sources. Here, we show that the nearly 150,000 warehouses in the U.S. worsen local traffic-related air pollution with an average near-warehouse $NO_2$ enhancement of nearly 20% and are disproportionately located in marginalized and minoritized communities. Near-warehouse truck traffic and $NO_2$ significantly increase as warehouse density and the number of warehouse loading docks and parking spaces increase. Increased satellite-observed $NO_2$ near warehouses underscores the need for indirect source rules, incentives for replacing old trucks, and corporate commitments towards electrification. Future ground-based monitoring campaigns may help track impacts of individual or small clusters of facilities.

In recent decades, globalization and e-commerce have fueled the need for warehouse receiving, sorting, and fulfillment facilities[1–6]. The transportation infrastructure needed to redistribute goods among these facilities and deliver them to retail centers or consumers is enormous. For example, in 2021, Amazon, an industry leader in retail e-commerce, operated around 175,000 delivery vans and more than 37,000 semi-trailers in the U.S., with increased fleet growth expected in the future.

Warehousing and goods movement increase traffic-related air pollutants, including nitrogen dioxide ($NO_2$) and fine particulate matter[1,7]. These pollutants are associated with pediatric asthma development, cardiovascular disease, premature death, and other health impacts[8] and disproportionately affect communities with relatively lower income and racial and ethnic minorities[9,10].

Concerns about air pollution around warehouses in Southern California spurred the South Coast Air Quality Management District to issue the Warehouse Indirect Source Rule, requiring that warehouses greater than 100,000 square feet directly reduce emissions of nitrogen oxides ($NO_X$) and diesel particulate matter[11]. Similar rulings are being considered in other U.S. states such as New York[12] and Colorado[13]. Yet, no study has quantitatively connected warehousing and air pollution or explored the potential environmental equity implications of warehousing on a nationwide basis. Shearston and colleagues[1] observed increased truck traffic and small increases in air and noise pollution following the opening of a warehouse in New York City. A recent report examining the warehousing industry in ten U.S. states determined approximately 15 million people live within approximately 1 km of warehouses, and the Black, Latino, Asian, and American Indian populations are overrepresented among these fenceline populations[2].

Here we provide a comprehensive nationwide study examining traffic, associated air pollution, and environmental equity near warehouses in the contiguous U.S. We colocated warehouse locations from

[1]Department of Environmental and Occupational Health, George Washington University, Washington, DC, USA. [2]International Council on Clean Transportation, Washington, DC, USA. ✉e-mail: gaigekerr@gwu.edu

a proprietary dataset, CoStar, with space-based measurements of the traffic-related air pollutant $NO_2$ from the TROPOspheric Monitoring Instrument (TROPOMI), high-resolution $NO_X$ emissions estimates, traffic data from the Highway Performance Monitoring System, and demographic information from the U.S. Census Bureau (Section 4). Results can enable policymakers at the local, state, and national levels to better understand and address the health-harming air pollution associated with this burgeoning industry.

## Results

As of 2021, 149,075 warehouses existed throughout the contiguous U.S., with nearly 20% of these warehouses located in just ten, or 0.3% of, counties: Los Angeles, California; Harris, Texas; Cook, Illinois; Miami-Dade, Florida; Maricopa, Arizona; San Bernardino, California; Orange, California; Dallas, Texas; Alameda, California; and Cuyahoga, Ohio (Figure S1). Using a per capita measure of warehouses to make direct comparisons across counties, we found a large number of warehouses per capita in major urban areas but also in some rural areas (Fig. 1A and Supplementary Data 1). Several of these rural areas are colocated with gas and oil extraction, and warehouses per capita in these areas can be greater than in highly urbanized counties (e.g., Williams County, North Dakota, near the Bakken formation, has 127 warehouses per 100,000 compared with 65 per 100,000 in Los Angeles County).

The number of new warehouses constructed annually has decreased over the last four decades, roughly consistent with U.S. domestic economic activity (Fig. 1B). Warehouse construction accelerated in the 2010s as the U.S. economy recovered from the 2007-2008 financial crisis with an 117% increase in the total number of warehouses built between 2021 and 2010. New warehouses built during this period were significantly larger in size, had a greater ability to handle traffic (as indicated by the significant increase in the number of loading docks and parking spaces), and tended to be built in census

tracts containing a significantly larger number of existing warehouses ("clustering"; Fig. 1C–F). Among the most dramatic changes between 2010 and 2021 was the 400% increase in the median number of loading docks at new warehouses.

We created a composite of near-warehouse satellite-derived $NO_2$ averaged over all warehouses in the continental U.S. in 2021 (Fig. 2A). The near-warehouse area was defined as the $\pm 7$ grid cells ($\sim 7$ km) extending in each cardinal direction from the warehouse using the TROPOMI $NO_2$ grid. To estimate the impact of warehouse-related activity on nearby $NO_2$ pollution, we calculated the warehouse-related $NO_2$ enhancement as the relative difference between wind direction-adjusted $NO_2$ averaged over the upwind orthogonal edge of the composite and the maximum $NO_2$ level in the composite, which is displaced $\sim 4$ km downwind of warehouses on average (Fig. 2A). Here, the upwind orthogonal edge refers to the fourteen grid cells on the upwind edge (left side) of the composite. Since the composite was formed with wind direction-adjusted TROPOMI measurements, we expect $NO_2$ levels averaged over this upwind orthogonal edge will minimize contributions from warehousing-related activities. With this approach we find that activities near warehouses are associated with a 17.9% increase in $NO_2$ on average across the continental U.S. The $\sim 4$ km displacement between $NO_X$ emission sources (i.e., warehouses) and peak $NO_2$ occurs for other $NO_X$ sources[14] and likely stems from most $NO_X$ (90–95%[15]) being emitted as NO and thereafter oxidizing to form $NO_2$ generally within minutes depending on available oxidants and meteorological conditions[16]. Although $NO_2$ distance-decay gradients are small ($<1$ km[17,18]), peak levels of satellite-measured $NO_2$ can occur several km downwind of $NO_X$ sources or source regions[19–21], consistent with our findings in Fig. 2A.

To determine whether the results from Fig. 2A are an artifact of warehouses being located in areas with higher baseline $NO_2$, we repeated the analysis but for warehouses in urban versus rural areas and different population density classifications ($<267$ people/

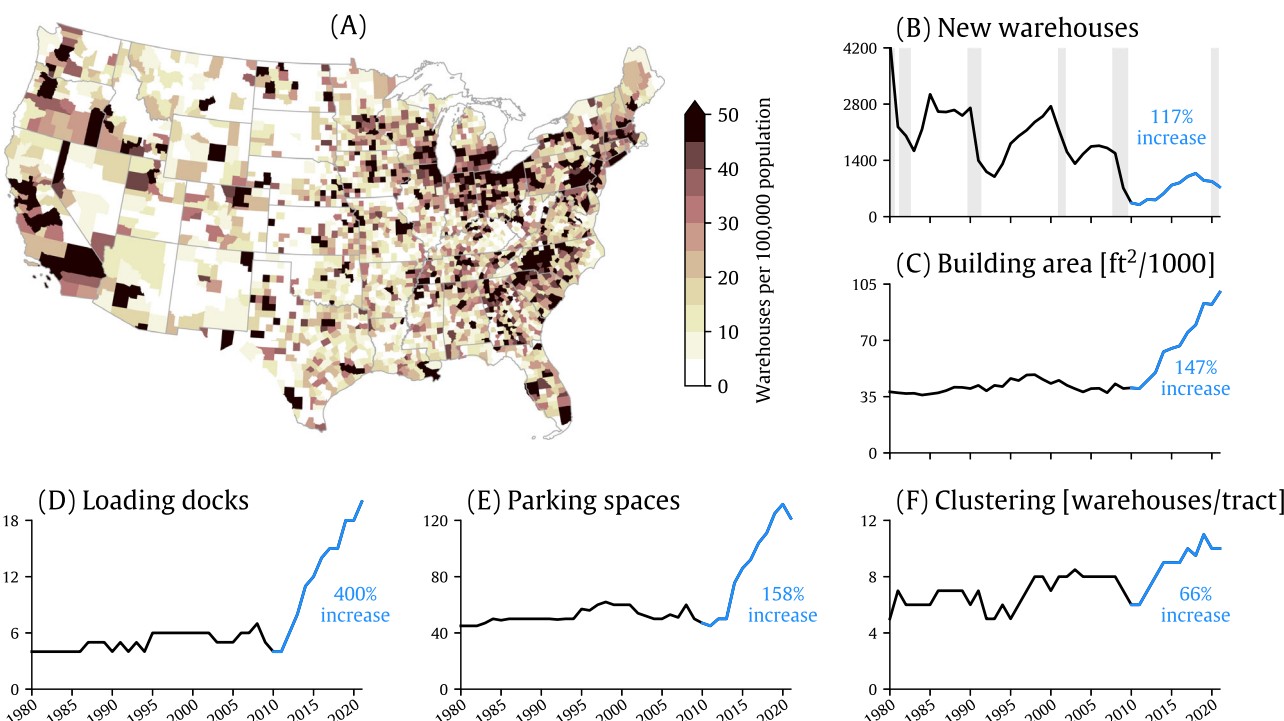

**Fig. 1 | Spatial distribution of U.S. warehouses and trends in warehouse characteristics. A** The number of warehouses per capita in U.S. counties as of 2021. Time series in (**B**–**F**) highlight trends in the total number of new warehouses constructed and warehouse characteristics (e.g., building area, loading docks, etc.), indicated by the median value of the characteristic for all warehouses built during the calendar year. The blue portion of the time series and corresponding text highlight relative changes between 2010 and 2021. Gray shading in (**B**) denotes the dates of U.S. recessions as inferred by a gross domestic product-based recession indicator[57].

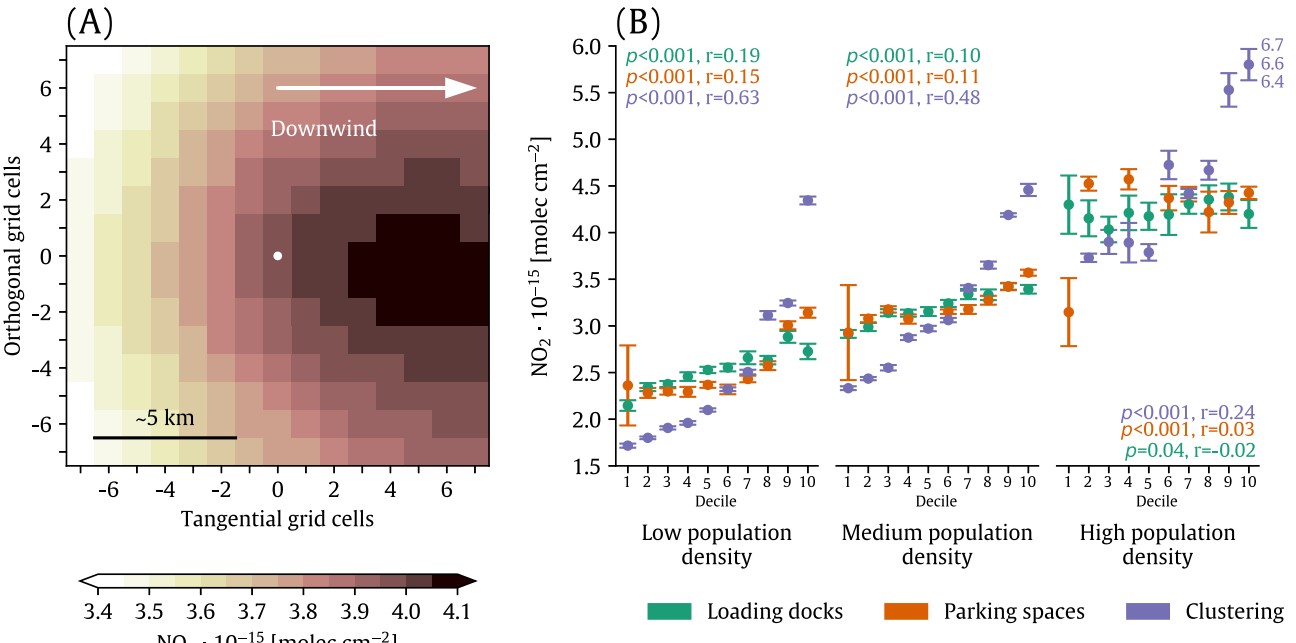

**Fig. 2 | Spatial variations and role of warehouse characteristics in near-warehouse NO₂ enhancements. A** Annual average 2021 TROPOMI NO₂ composited over all warehouses in the contiguous U.S. The white scatterpoint corresponds to the location of the warehouses, and the ±7 grid cell buffer around the warehouse represents our definition of the "near warehouse" environment. **B** Median NO₂ levels (scatterpoints) and the 95% confidence interval (vertical bars) formed by discretizing warehouses that comprise the composite shown in (**A**) into deciles based on their number of loading docks, parking spaces, and the number of warehouses in each census tract ("clustering") for different population density classifications. Inset text indicates the *p*-value (*p*) calculated using a two-sided Wald test with a *t*-distribution of the test statistic and the Spearman's rank correlation coefficient (*r*) of the relationship between NO₂ and property characteristics but calculated using the full dataset, rather than the deciles. Median NO₂ levels and the associated confidence interval for the top clustering decile in high population density areas are out of the frame and are shown alongside the plot.

km² = Low density, 267–1501 people/km² = Medium density, and >1501 people/km² = High density). Near-warehouse NO₂ increases persist across these environments, ranging from 4.3-21.5% (Figure S2). Warehouses are increasingly constructed in close proximity to other warehouses (Fig. 1F). The NO₂ enhancement observed near warehouses in Fig. 2A does not account for this clustering, so we also partitioned warehouses based on the level of clustering and found, on average, enhanced NO₂ for even single warehouses (Figure S3). We note, however, that the magnitude of the average NO₂ enhancement for single warehouses (i.e., 11.5%; Figure S3A) is smaller than the enhancement for all warehouses regardless of clustering (i.e., 17.9%; Fig. 2A) or for the largest clusters of warehouses, which have average enhancements of 23.1% (Figure S3J).

We explored whether near-warehouse NO₂ varies with warehouse characteristics we expect to be causally linked to NO₂: warehouse density and their ability to handle traffic. The number of warehouse loading docks and parking spaces–proxies for warehouse traffic–and warehouse clustering are associated with significant, nearly monotonic NO₂ increases across low and medium-density environments (Fig. 2B). Clustering exhibits the strongest association with NO₂, explaining nearly 40% of the NO₂ variance near warehouses sited in low population density environments. The relationship between NO₂ and loading docks, parking spaces, and clustering is weaker in high population density environments, likely owing to the complex mixture of NOₓ sources in built-up areas that may obscure the warehouse-associated NO₂ enhancement. These associations continue to hold when restricting the size of the area defined as "near the warehouse" (Figure S4). Other warehouse characteristics were either uncorrelated or only weakly correlated with NO₂; for example, the correlation between warehouse building area and NO₂ is weak for all population density environments (*r* < 0.1).

In addition to showing how traffic proxies at warehouses relate to increased NO₂, we also explored how observed truck traffic relates to near-warehouse NO₂ and warehouse characteristics. We analyzed the relationship between truck traffic, characterized by truck vehicle kilometers traveled (VKT), and warehouse characteristics for warehouses binned by total (i.e., all vehicles) VKT since truck VKT scales with the total VKT (Figure S5). Within these bins, larger warehouse clusters and more loading docks and parking spaces were generally associated with increased truck traffic near warehouses (Fig. 3A and Figure S5). The precise increase of truck traffic and related significance varied based on the total traffic, but averaged across the decile bins, an additional loading dock and warehouse within the same census tract were associated with a 1456 and 485 km increase of truck VKT within the ± 7 grid cell near-warehouse area, while an additional parking space was associated with a 13 km decrease in truck VKT. Warehouse characteristics and truck VKT near warehouses were negatively or not significantly associated in areas with the highest total VKT, which are predominantly in urban cores (Figure S6). Our definition of truck traffic includes both long- and short-haul trucks, and the negative relationship between truck traffic and warehouses in urban areas might reflect that long-haul truck activity may be restricted in cities where local deliveries from short-haul trucks are allowed. In addition, navigational rerouting to avoid congestion might contribute to this negative relationship as well. Near-warehouse NO₂ increased significantly as near-warehouse truck traffic increased across different population density environments (Fig. 3B). These results are robust when reducing the near-warehouse area extent, although the precise increase in truck VKT given increases in clustering, loading docks, or parking spaces slightly decreases when we reduce the size of the near-warehouse area (Figure S7). Since the analysis in Figure S7 considers total VKT summed over near-warehouse definitions with decreasing

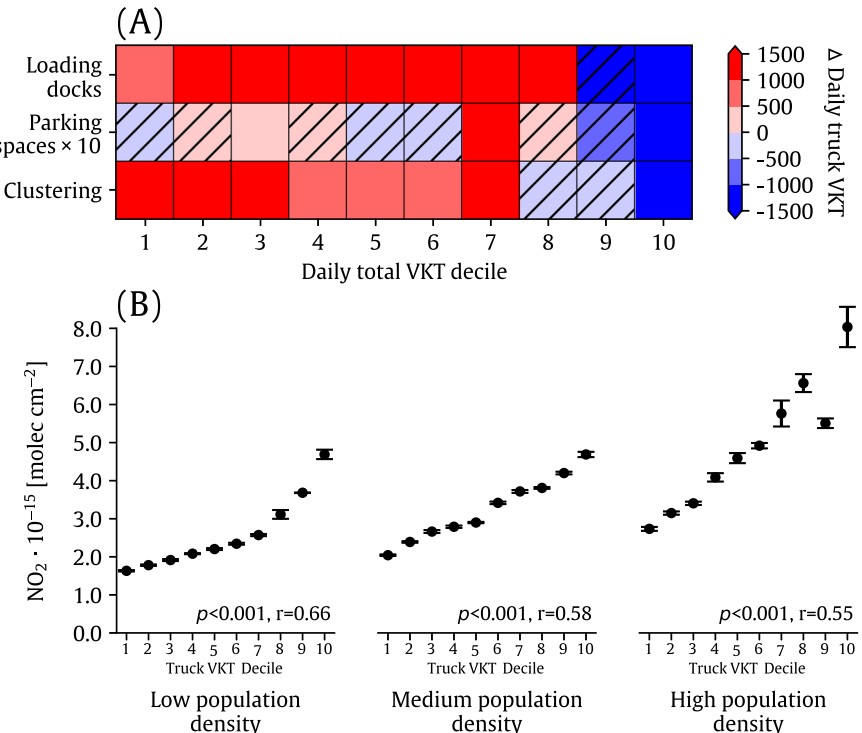

**Fig. 3 | Relationships between warehouse characteristics and NO$_2$ with truck traffic. A** Slope of the linear regression between daily truck vehicle kilometers traveled (VKT) near warehouses versus the number of loading docks, the number of parking spaces scaled by 10, and clustering for decile bins of daily total VKT near warehouses. Hatching represents a bin where the relationship between daily truck VKT and a particular warehouse characteristic is not statistically significant. The distributions from which the slopes are derived are shown in Figure S5. **B** Same interpretation and statistical analysis as Fig. 2B, but near-warehouse NO$_2$ levels are binned by near-warehouse daily truck VKT for low, medium, and high population density environments.

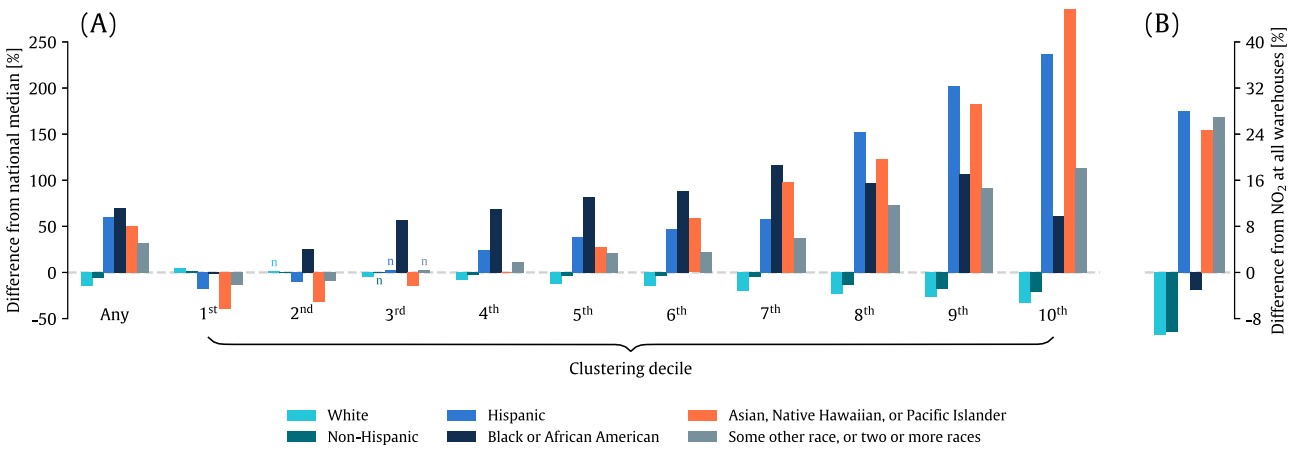

**Fig. 4 | Disparities in the racial-ethnic composition of the near-warehouse population. A** The relative difference in the racial and ethnic composition of census tracts containing warehouses and the overall U.S. median demographics. This difference is calculated for all tracts containing a warehouse or multiple warehouses ("Any") and tracts discretized into deciles corresponding to the number of warehouses in each tract ("Clustering decile"). Bars marked with "n" denote differences between demographics at warehouses and national median demographics that are not statistically significant using a two-sample Kolmogorov-Smirnov Test. **B** The relative difference in population-weighted NO$_2$ averaged near warehouses for different population subgroups and the overall population-weighted NO$_2$ near warehouses.

areal extents, the decrease in truck VKT with these decreasing extents likely stems from a smaller amount of total traffic when the near-warehouse definition is restricted.

NO$_X$ sources and NO$_2$ levels are inequitably distributed among different racial-ethnic groups[10,22–24], and we investigated whether warehouses and associated NO$_2$ were consistent with these previous findings. Compared with national median demographics, tracts with warehouse(s) had a larger Hispanic population (by 59.7%), Black or African American population (by 69.5%); Asian, Native Hawaiian, or Pacific Islander population (by 50.9%); and population identifying as some other race or two or more races (by 31.5%) in 2021 (Fig. 4A). As the number of clustered warehouses increased, the proportions of racial-ethnic minorities also increased. In the top ten percent of tracts with the most warehouses, the proportion of the Hispanic population was nearly 240% higher (28.1% versus 8.3%) and the Asian population nearly 290% higher than U.S. median values (6.2% versus 1.6%).

Increasing parking spaces, loading docks, and building areas were generally linked with a larger proportion of racial-ethnic minorities, although not for every population subgroup; for example, the proportion of the Black or African Americans population increased with additional loading docks and greater building area but decreased with additional parking spaces (Figure S8). The nationwide disparities in Fig. 4A largely hold for individual U.S. states, although the proportion of racial-ethnic minorities in tracts with warehouse(s) is smaller than the state median in a few states (e.g., Idaho, Montana; Figure S9). These states are largely rural and historically tend to have fewer racial-ethnic minority populations, which both could contribute to these findings. We encourage future, state-specific studies that provide further context to demographic patterns of warehouse siting.

The inequitable distribution of warehouse and warehouse characteristics, which we previously linked to increased $NO_2$, results in higher $NO_2$ for racial-ethnic minorities living near warehouses relative to overall near-warehouse $NO_2$ levels (Fig. 4B). The Hispanic population and the population identifying as some other race or two or more races experience the largest difference relative to the overall near-warehouse $NO_2$ (by 29.1% and 28.0%, respectively). Despite the Black or African American population living in close proximity to a greater number of warehouses and associated loading docks and parking spaces (Fig. 4A and Figure S8), this population group, on average across the U.S., faces lower $NO_2$ by 2.1% than average near-warehouse levels. However, the near-warehouse Black or African American population is exposed to higher near-warehouse $NO_2$ than the overall near-warehouse population within 43 of the 49 states comprising the contiguous U.S. (Figure S10). If we remove these six states from our analysis and recalculate the relative difference in $NO_2$ for the Black or African American population relative to the overall near-warehouse $NO_2$, we find that this population subgroup faces slightly higher $NO_2$ than average near-warehouse levels (0.8%).

While we demonstrated the association among warehouses, traffic, and traffic-related pollution, exogenous $NO_x$-emitting activities could also influence near-warehouse $NO_2$. To understand the impact that other emitting sectors might contribute to the TROPOMI $NO_2$ signal, we examined whether the signal persists across different ranges of emission intensities and different dominant emitting sectors using a high spatial resolution emissions dataset, the Neighborhood Emission Mapping Operation (NEMO). We find enhanced $NO_2$ near warehouses regardless of underlying levels of emissions and dominant emitting sectors (Figure S11), indicating that our results are not an artifact of warehouses being colocated with other $NO_x$ sources. The exact enhancement varies depending on underlying emissions and sector but ranges from ~5–30%, with smaller enhancements generally observed near warehouses located in areas with lower-intensity emissions. While CoStar does not provide information on warehouse type or purpose, partitioning warehouses by underlying emission intensity might serve as a proxy. For example, the subset of warehouses located in areas with low on-road emissions (Figure S11A) might represent facilities intended for long-term storage warehouses rather than heavily trafficked last-mile facilities and, accordingly, we observed a smaller $NO_2$ enhancement at these facilities.

## Discussion

This nationwide study substantiates growing concerns from fenceline communities and policymakers of increased truck traffic and traffic-related air pollution near warehouses. Local $NO_2$ enhancements near warehouses are nearly 20% on average across the continental U.S. Given the early afternoon overpass of TROPOMI coupled with a mid-morning peak in heavy-duty vehicle activity[25], it is possible that this enhancement may be an underestimate. Warehouses with more loading docks and parking spaces–typical of warehouses built during the 2010s–attract the most traffic and are associated with the highest $NO_2$ levels. Warehouses are also concentrated in communities with large racial-ethnic minority populations, likely compounding the existing environmental, health, and socioeconomic burdens these populations experience. Our study also demonstrates that satellite observations of $NO_2$ can identify warehouse-related pollution enhancements when aggregated at state and national levels.

This study builds on several others that have documented higher $NO_2$ and associated public health damages for marginalized populations[26–28]. Previous work has also explored how specific sectors contribute to disparities in $NO_2$ and other pollutants, finding inequities stemming from heavy-duty diesel vehicles[22,23], power generation and point sources[29], and freight transport[30]. The warehousing industry–related to both heavy-duty and freight transport–is yet another industry that contributes to air inequality that we uncovered through this study. The small magnitude of the population-weighted near-warehouse $NO_2$ level for the Black or African American population relative to the total population-weighted level (Fig. 4B) was one unexpected finding of this study and could stem from the large Black or African American populations in the southeastern U.S., where $NO_2$ concentrations are generally lower.

Future work might use photochemical models to conduct source apportionment to understand the fraction of near-warehouse pollution stemming specifically from warehousing-related activity and provide surface-level pollutant concentrations that enable warehousing-related health impact assessments. At the moment, however, this type of modeling is hamstrung for two key reasons. First, the highest spatial resolution for nationwide model simulations is typically > 10 km x 10 km[31,32], which is several times larger than $NO_2$ distance-decay gradients[27]. Second, emission simulators (e.g., MOVES[33]) only recently added the capability to spatially allocate off-network idling emissions at warehouses. At present, satellite data represent the most empirical means to surveil $NO_2$ with complete spatial coverage and has been shown to capture observed surface-level spatial variations in $NO_2$ concentrations[34]. The relationships and warehouse-related $NO_2$ proxies (e.g., loading docks and parking spaces) reported in our results present an opportunity to further develop these emission simulators to account for warehousing activities. Future work might also leverage the recent launch of the Tropospheric Emissions: Monitoring of Pollution (TEMPO) satellite. TEMPO's ability to surveil $NO_2$ throughout daytime hours has the potential to increase our understanding of warehousing impacts on air quality and test whether the previously discussed hypothesis that TROPOMI underestimates $NO_2$ on account of its overpass time and diurnal trends in heavy-duty traffic. The deployment of regulatory and/or low-cost air quality surface-level monitors near existing or planned warehouses[1] is another exciting area of future research that would complement space-based observations.

Since some warehouses are missing data on loading docks and parking spaces (Section 4.1), we tested whether these facilities are correlated with other variables included in our study in such a way that could bias results. We found that the correlation between missing warehouse parking spots or loading dock data with TROPOMI $NO_2$, race-ethnicity data, population density, and truck VKT was generally $\pm 0.1$ (the correlation between truck VKT and missing warehouse parking spots was the exception, $r = -0.11$). Given that there is little correspondence with missing characteristics and pollution, demographics, and traffic, it is unlikely that the incomplete warehouse data would systematically bias our key results. Our analysis linking warehouse characteristics with truck traffic (Fig. 3A) assumes linear relationships. We explored whether this assumption was justified and found normally-distributed residuals (Figure S12), supporting this linear approach. We also note that the interpretation of the linear approach (i.e., change in truck traffic per unit change in loading docks, parking spaces, or clustering) is easily interpretable compared with more complex linear or non-linear models. One other potential limitation is our use of tropospheric column measurements of $NO_2$.

There is a potential that $NO_2$ may be present aloft rather than at the surface if a plume is transported from a large emitter; however, we believe the ability for this potential to substantially influence our results is minimal, given the strong correlation observed between columnar and surface-level concentrations[14].

Since ~2000, consumer reliance on e-commerce, and therefore warehousing, has increased[35], especially during the COVID-19 pandemic[36]. Our results linking warehousing with increased traffic-related pollution represent 2021, at which point commercial trucking activity near warehouses had not only rebounded from but increased since the start of the COVID-19 pandemic[37]. The results come during a critical window of opportunity to inform zoning and transportation policy, environmental impact statements, and emissions standards to protect public health and equity. Federal emission standards tightening $NO_X$ emission limits for new vehicles starting in 2027[38] will help reduce average $NO_X$ emissions; however, given the aforementioned trend of increasing e-commerce[35], additional state and local policies and corporate action will also be important for reducing $NO_2$. At the state level, state energy commissions could direct utilities to devise plans to expedite permitting of charging stations in and around warehouses[39], and indirect source rules may help regulate and offset warehouse-related $NO_2$. Local actors may implement low-emission zones or environmental zones that target in-use emission reductions in specific areas[40]. Corporate action replacing older trucks and commitments towards electrification may further protect public health and address inequities. Further research is needed to determine the utility of satellite $NO_2$ data for evaluating the impact of individual facilities or clusters in limited geographical areas.

## Methods
### Warehouse locations and characteristics
Information on warehousing comes from CoStar, which relies on staff, aerial, drone, field research, and public records to populate its database, which currently contains over 6.9 million commercial real estate properties. We searched CoStar for warehouses in the contiguous U.S. built through 2021 (CoStar filters: Construction Status = Existing, Property Type = Industrial, Secondary Type = Warehouse), which yielded ~470,000 properties. We ultimately chose to only consider warehouses with a building area >20,000 ft$^2$ as contemporaneous studies have also considered only larger warehouses[41], and CoStar's limit of 500 properties per download would have resulted in a large number of separate queries and downloads. CoStar provides data on warehouse characteristics such as their square footage, number of parking spaces, number of loading docks, and year built that we use in this study. Here, loading docks refer to stalls level with heavy-duty truck trailers that allow inventory from the trucks to be loaded or unloaded, while parking spaces correspond to surface-level stalls generally intended as parking for light- or medium-duty vehicles. Some of the warehouses are missing data on these characteristics, such as the year built (9.9%), the number of loading docks (40.7%), and the number of parking spaces (31.1%).

CoStar does not provide information on quality control or assurance, and it is difficult to verify the spatial distribution and characteristics of warehouses from CoStar as we know of no other comprehensive dataset—either publicly available or proprietary—that provides information on warehousing. We note, however, that CoStar is the most extensive commercial real estate database in the U.S., undergoes vetting and updates by their team of researchers, and is commonly used in academic research[36,42,43].

### Remotely-sensed $NO_2$
TROPOMI (version 02.03.01[44]) measures tropospheric column $NO_2$ at an unprecedented native resolution of 3.5 km x 5 km with an early afternoon (~1330 h local time) overpass time. We obtained daily TROPOMI retrievals over the contiguous U.S. from 2021 and increased their spatial resolution to 0.01° x 0.01° (~1 km x 1 km) through oversampling, discarding retrievals with quality assurance values lower than 0.75[14]. We sampled daily oversampled TROPOMI $NO_2$ at the grid cell nearest to each warehouse and ±7 grid cells (approximately ±7 km) in each cardinal direction, referring to this ±7 grid cell extent as "near warehouse." While the choice of the ±7 grid cell buffer is arbitrary, we found that it captures the near-warehouse $NO_2$ plume displaced slightly downwind of the warehouse (Fig. 2A). We also tested how other near warehouse definitions impact our results by restricting the buffer to smaller extents around warehouses (i.e., ±5 grid cells, ±3 grid cells, and ±1 grid cells) but found that our key results do not hinge on the precise definition of "near warehouse" (Figure S4).

Variations in the prevailing wind direction could obfuscate $NO_2$ enhancements near warehouses when daily satellite-derived $NO_2$ near warehouses is composited. As a result, we obtained hourly 0.25° x 0.25° 100-m wind data for 2021 from the European Center for Medium-Range Weather Forecasts fifth-generation reanalysis[45]. Reanalyzed winds at this altitude have been evaluated on hourly timescales against in-situ observations (largely in the U.S. and Europe) and were found to capture observed values with a mean absolute error of 9–13%[46]. We sampled and averaged these hourly fields between 16-21 UTC, which is approximately centered around the overpass time of TROPOMI over the continental U.S, and applied bilinear interpolation to increase the spatial resolution to 0.01° × 0.01° for direct comparison with the oversampled TROPOMI data[47]. We found the mean wind direction near each warehouse following Yamartino[48] using the arctangent of the zonal and meridional wind and artificially rotated each daily $NO_2$ footprint tangential to the prevailing wind (Figure S13). Finally, we averaged all daily, rotated $NO_2$ at each warehouse in 2021 to account for seasonal variations in $NO_2$[14]. While TROPOMI does not measure surface-level $NO_2$, it exhibits a quasi-linear relationship and high correlation with surface-level observations[14,23] and has enabled the detection of individual point sources and intraurban variations of $NO_2$[19,20,22,23,27,49].

Multiple warehouses can be clustered together within the same ±7 grid cell buffer or even within the same 0.01° x 0.01° TROPOMI grid cell. It is not possible to disentangle which fraction of satellite-observed $NO_2$ might be attributable to a given warehouse in a cluster. Accordingly, some of our results present $NO_2$ enhancements near warehouses in the aggregate; that is, not controlling for multiple colocated warehouses (e.g., Fig. 2A). We also partition warehouses into groups based on the level of clustering (e.g., Fig. 2B and Figure S3–4) to understand average $NO_2$ enhancements for single versus clusters of warehouses.

### $NO_X$ emissions
To understand how the satellite-measured $NO_2$ signal near warehouses might be influenced by underlying emissions, we considered a 1 km x 1 km emissions dataset, NEMO[50]. Briefly, this dataset is based on the U.S. Environmental Protection Agency's National Emission Inventories 2017 (NEI 2017) with additional fine-scale spatial allocation using over a hundred spatial surrogates such as population, traffic counts, and oil and gas wells. In addition to providing total $NO_X$ emissions, NEMO also provides sector-specific emissions from nine sectors: anthropogenic fugitive dust emissions (henceforth "dust"), agricultural ammonia sources, nonpoint sources not in other sectors ("non-point"), nonpoint oil and gas-production-related sources ("oil and gas"), locomotive sources on railroads ("rail"), residential wood combustion sources ("residential wood combustion"), on-land mobile sources that drive on roads ("on-road"), On-land mobile sources not on roads or railroads ("non-road"), and Airport emissions ("airports"). Because agricultural ammonia sources were not associated with any NEMO $NO_X$ emissions across the U.S., we did not further consider emissions from this sector in our analysis.

Similar to our analysis of TROPOMI $NO_2$, we sampled the NEMO $NO_X$ emissions at the grid cell nearest to each warehouse and ± 7 grid cells in each cardinal direction; however, since these emissions represent a surface flux, unlike the column-integrated TROPOMI $NO_2$ levels that are prone to prevailing wind patterns, we did not rotate the NEMO $NO_X$ emissions in the direction of the prevailing wind. Matching these 2017 NEMO $NO_X$ emissions with 2021 TROPOMI $NO_2$ is not a perfect comparison, but we are not aware of a more recent emissions dataset at a resolution commensurate with TROPOMI. Moreover, changes in land use (e.g., road density and volume, urban form) are less likely to drastically change in a developed country such as the U.S. compared with developing areas[51].

## Demographic data

Demographic data at the census tract level were derived from the American Community Survey (ACS) 5-year estimates spanning 2017–2021[52]. The ACS is conducted by the U.S. Census Bureau in the years between the decennial censuses and provides statistics that are controlled to match the Bureau's annual population estimates by age, sex, ethnicity, and race[53]. The 5-year estimates are considered the most reliable ACS product and represent data from the entire period, the final year of which (2021) is the primary year of our analysis. We paired the tract-level demographics with corresponding tract geographic boundaries from the 2020 Census[54] and assessed the racial-ethnic composition of the population living in the same census tracts in which warehouses are located using six different categories: white; Non-Hispanic; Hispanic; Black or African American; Asian, Native Hawaiian, or Pacific Islander; and some other race or two or more races. These categories are non-mutually exclusive since individuals can, for example, identify as white and non-Hispanic or Black and Hispanic. We use census tract total population estimates and the area of each tract to calculate the population density and assign each warehouse a population density based on the census tract in which the warehouse is located. To assess race- and ethnicity-specific near-warehouse $NO_2$ levels, we calculate the population-weighted $NO_2$ near warehouses with the following:

$$NO_{2,population-weighted} = \frac{\sum_{i=1}^{n}(NO_{2,i}\,pop_i)}{\sum_{i=1}^{n}pop_i} \qquad (1)$$

Here, $NO_{2,population-weighted}$ represents population-weighted near-warehouse $NO_2$ for the total population or a particular racial-ethnic population group, $i$ corresponds to an individual warehouse, $NO_2$ is spatially-averaged $NO_2$ levels near (± 7 grid cell) warehouse $i$, and $pop$ represents the total population or the population of a particular racial-ethnic population group in the census tract in which warehouse $i$ is located. We then calculate the percent difference between $NO_{2,population-weighted}$ for a particular racial-ethnic population group (i.e., white, non-Hispanic, Black, or African American, etc.) and $NO_{2,population-weighted}$ for the total population to quantify whether certain population groups living near warehouses experience worse traffic-related air pollution than overall near-warehouse levels.

## Traffic data

We incorporated traffic data from the Highway Performance Monitoring System (HPMS), which provides annual average daily traffic (AADT) by vehicle class for 14.3 million road segments representing highways, collector roads, and arterial roads in the contiguous U.S. At the time of this study's completion, only HPMS data through 2020 were publicly available. Because traffic in 2020 was affected by the COVID-19 pandemic, we paired the 2019 HPMS dataset with the other data inputs from 2021, acknowledging that traffic volume and patterns may have changed between 2019 and 2021. We formed two different measures of AADT: total AADT and truck AADT (i.e., AADT from vehicle classes 4–13, which include single-unit trucks, buses, and semi- and multi-trailers).

HPMS synthesizes data from individual U.S. states' departments of transportation, leading to variations in the spatial completeness of the data. To address this limitation, each state's AADT was scaled using the ratio of total activity reported by HPMS to the total activity reported by the nationally consistent Federal Highway Administration database[55]. We then transformed total and truck AADT on individual road segments to vehicle kilometers traveled (VKT = AADT * road length) due to the variable lengths of road segments and thereafter summed total and truck VKT over all road segments contained within 0.01° x 0.01° grid cells over the contiguous U.S. (Figure S14). For every warehouse, we sampled the 0.01° x 0.01° traffic grid at the grid cell nearest to each warehouse and ± 7 grid cells in each direction and summed total and truck VKT over this area to represent near warehouse traffic, similar to our treatment of TROPOMI data.

## Statistical analysis

We measured whether relationships exist among warehouse characteristics, $NO_2$, and traffic using the slope of the least squares regression and the nonparametric Spearman's rank correlation coefficient ($r$). The significance of these relationships was quantified using the Wald test with a $t$-distribution of the test statistic and a significance threshold of 0.05. We assessed the significance of temporal trends using the non-parametric Mann-Kendall test, declaring trends as significant if the significance level associated with the test is below 0.05. When determining whether demographics in census tracts containing warehouses significantly differed from national or state demographics, we employed the two-sample Kolmogorov-Smirnov Test, again with a significance threshold of 0.05.

When examining the relationship between warehouse characteristics and $NO_2$, we partitioned warehouses by 2021 tract-level population density tertiles in the contiguous U.S. (< 267 people/$km^2$ = Low density, 267–1501 people/$km^2$ = Medium density, and > 1501 people/$km^2$ = High density) to disentangle changes in $NO_2$ associated with warehouse characteristics from enhancements due to different levels of urban activity. In a similar vein, we quantified associations between warehousing and traffic by partitioning warehouses based on daily total VKT to effectively control for different baseline traffic volumes before calculating the change in daily truck VKT.

In visualizations, we present results grouped by deciles (median values and the 95% confidence interval for the distribution of each decile) for interpretability, but we note that measures of significance and correlation related to these visualizations were calculated using the full distributions of particular variables. We did not formally assess significance for population-weighted near-warehouse $NO_2$ levels among different racial-ethnic population subgroups (Section 4.4, Fig. 3B) given that this comparison is between pairs of singular values rather than between distributions.

## Reporting summary

Further information on research design is available in the Nature Portfolio Reporting Summary linked to this article.

## Data availability

The locations and characteristics of U.S. warehouses are proprietary but are available for purchase from CoStar (https://www.costar.com). Other raw data used in our study are publicly available using the following: TROPOMI $NO_2$ (version 02.03.01, https://doi.org/10.5270/S5P-9bnp8q8) via the Copernicus Data Space Ecosystem at https://dataspace.copernicus.eu/; U.S. Census Bureau ACS estimates at https://www.nhgis.org; NEMO emissions at http://air.csiss.gmu.edu/aq/NEMO/index.html; and the traffic data at https://www.fhwa.dot.gov/policyinformation/hpms.cfm. Post-processed datasets are available on our study's Zenodo repository: https://doi.org/10.5281/zenodo.11577869.

## Code availability

Code supporting statistical analyses and visualizations are publicly available on our study's Zenodo repository: https://doi.org/10.5281/zenodo.11577869.

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

## Acknowledgements
This work was completed, in part, with resources provided by the High Performance Computing Cluster at George Washington University, Information Technology, Research Technology Services[56]. We are grateful to Kevin Bryant and Jonathan Jump for their assistance navigating and interpreting CoStar data products. Funding for this work was provided by NASA (grant no. 80NSSC21K0511) to SCA and DLG.

## Author contributions
GHK conceived and designed the research, led the analysis, and wrote the paper. G.H.K., D.L.G., and M.M. acquired and analyzed the data. S.C.A. and J.M. were responsible for funding acquisition. All authors reviewed and interpreted results and contributed to editing the paper.

## Competing interests
GHK reports serving as a consultant for the Environmental Defense Fund, the Department of Justice, and the California Air Resources Board. SCA reports serving as a consultant for the Environmental Defense Fund, Department of Justice, and Environmental Integrity Project. The remaining authors report no competing interests.
