## [Peer Review File · Nature Communications]

Air pollution impacts from warehousing in the United States uncovered with satellite dataREVIEWER COMMENTS

Reviewer #1 (Remarks to the Author):

Summary

This exploratory study assesses correlations between cross-sectional satellite measured NO₂ concentrations, traffic and warehousing across the United States, and whether warehouse locations are distributed unequally among different demographic groups. Using straightforward spatial statistics, the authors find that warehouses are correlated with elevated NO₂ concentrations and disproportionate exposures on non-white populations. I have no major comments for the manuscript. The authors clearly describe the major limitations of their analyses and provide reasonable justifications for their findings. I provide some minor comments to improve clarity.

Introduction

- Please provide a citation for the statement made on lines 32-33 "warehousing and goods movement...fine particulate matter."

Main

- It would be helpful to explain why NO₂ concentrations increase almost exponentially between the 8th and 10th deciles for clustering in low population density warehouse locations.
- Please explain the type of parking spaces that are available at the warehouses. Are they for trucks or for employees?

Methods

- Explain why they chose a 7km buffer to examine warehouse-related NO₂
- With clustering of warehouses, how did the authors control for buffers for multiple warehouses located within the same buffer? Please explain.

Discussion:

- Please provide additional data needs and knowledge gaps that future research can fill to answer questions about the association between warehousing and air pollution, as well as environmental justice implications of warehousing.
- Please describe whether or how warehousing activities changed during the COVID-19 (i.e., the time period for this study) so that readers can understand the generalizability of the study results to present day.

Methods:

- Please describe how or whether CoStar data undergo any QA/QC, whether this dataset is comprehensive and any other potential limitations of the dataset that might bias the results.
- Please describe any limitations in the methods used to approximate NO₂ concentrations by combining them with wind data.

Supplementary File

- Please provide a full description of each figure so that they can be standalone figures. E.g., change "Same as Figure 1A in the main text..." to "The number of warehouses per per country in U.S. counties as of 2021"
- Figure s4: please add the decile number below each bin

Reviewer #2 (Remarks to the Author):

General Comment:

This paper examines the air pollution (using NO₂ as a proxy) and truck traffic impacts of warehouses across the United States. The authors report that warehouses enhance local NO₂ levels by ~20% on average and that these impacts are disproportionately shared by marginalized communities.

This is a very important and timely paper. However, (1) several methodological approaches are not clearly described and, more importantly (2) for the implications of this paper and the strong conclusions that the authors draw, I believe more rigorous analyses are needed.

Below please find my specific comments.

1. Abstract "[...] an average near-warehouse NO₂ enhancement of nearly 20% and are disproportionately located in communities with above-average shares of Black, [...]": According to Figure 4B Black residents are exposed to lower NO₂ on average.
2. Abstract "Satellite data highlight the need for [...]": This is slightly awkwardly phrased. I understand that the authors here would like to highlight the advantage of using satellite data, but the way this is phrased it takes away something from the importance of the need that the findings highlight. Maybe rephrase?
3. Why/how was 2010 selected as the baseline year?
4. Figures 1C-F: Other than just visually, were these changes somehow statistically examined?
5. When reading this, I first thought that the +/- 7km buffer meant a circle with a 7-km radius around the warehouse. However, looking at the graphs, I realize that this approach includes in the near-warehouse zone those grid cells a part of which (the center? Some proportion?) falls within the 7-km buffer, resulting in a square (and not a circle). Please clarify.
6. How did the authors treat impacts of warehouses with overlapping buffers? How often did that occur?
7. How large was the area of the "upwind orthogonal edge of the composite"? Just 14 by 1 km²?
8. "In our case, the maximum levels were displaced ~4 km downwind of warehouses.": (a) On average? (b) What does "in our case" mean here?
9. Page 4 towards the end of the page (2-4 lines from the end): But wouldn't 4 km be too long for this process? NO₂ decay occurs at much shorter distances than that (as the authors also note in the discussion).
10. Figure 2: Why not transform NO₂ levels to ppm and report those? Wouldn't ppm be more relevant towards informing policy since standards are set using ppm?
11. Figure 2B: From what analyses are these p-values extracted? Was that a regression of e.g., NO₂ ~ loading docks in the lowest population density tertile?
12. The sensitivity analyses restricting near-warehouse zones to smaller buffers is not described in the Methods. Please add.
13. Figure 3A assumes linear associations. Were they linear indeed?
14. Last sentence of page 8: So six states make all the difference for warehouse-related NO₂ levels among Black residents? Were these the same states in which the 20% of the warehouses is located?
15. First sentence in the Discussion: This is just confounding. Could the authors repeat analyses adjusting for factors they think would confound these associations to obtain less biased estimates?
16. If I understood correctly, only areas with warehouses were included in analyses, yes? And the only comparison was with that upwind orthogonal edge? Why instead not match locations with warehouses to locations without warehouses but similar characteristics otherwise (e.g., population density, land use, building density, km of roads/highways, etc) to compare NO₂ levels? Are the upwind orthogonal edges exchangeable to the buffers in terms of potential confounders? If there are any systematic differences, then the reported results could be substantially biased.
17. There are many warehouses missing data on loading docks and number of parking spots. How were these warehouses treated in the analysis? Were they excluded? And was missingness informative based on some key characteristics (e.g., population density or land use)? If so, this could also substantially bias results.
18. I believe the TROPOMI overpass time is sometime mid-day, is that true? If so, how could that

impact the results estimated here? I would assume that warehouse-related traffic would peak overnight or early morning. Could then using TROPOMI potentially result in underestimation?

19. Section 4.3, last two sentences: I am sorry, but I cannot understand this analysis... Maybe include an equation to clarify? But also, how was significance evaluated in this analysis? Or was it not evaluated? This also has important implications for interpretation of results.

20. Limitations?

Reviewer #3 (Remarks to the Author):

Review of the manuscript "Satellite data uncover nationwide air pollution impacts from warehousing" by Kerr et al.

The manuscript describes an approach to relate satellite-based nitrogen dioxide observations (indicator of air pollution) near warehouses in the contiguous USA to warehouse-related parameters such as number of loading docks, parking spaces and clustering. In addition, the author analyzes the demographic composition near the US warehouses to assess how the near-warehouse NO₂ levels are distributed among different ethno-racial groups. The quality of the data used is appropriate for such analysis and the methods applied are proper and mostly support the conclusions. The results are of significance as warehouse-related activities are on the rise in recent years and warehouse-related activities and traffic-related air pollution might have substantial effects on local population, especially from certain ethno-racial groups in the country. There are nevertheless some weaknesses in my opinion in the way the data and results support the conclusions.

Specific comments:

- Title and intro: I think that the fact that analysis focuses on the US should be specified in the title. "nationwide" could refer to any "nation" unless you have a very USA-centric view of the world. The same applies to the introduction (third paragraph) where the authors use the word "states" twice to refer to US states. Please specify this more clearly.
- Sect. 2 first paragraph: You mention that warehouses in rural areas are often collocated with oil exploration and production facilities. Could then the NO₂-signal in such situations be mixed with possible emissions from oil extraction/processing activities? How do you make sure these signals are distinguishable? I suppose the authors assumed that the rotation procedure sufficiently isolates the contribution from different sources, but what happens when other sources are within or close to the near-warehouse definition of 7km? Could you provide for example a plot of the rotated and not-rotated NO₂-map for Bakken, and explain how/whether those signal overlap and how you treat such cases in your analysis?
- Related to the previous point, you mention that in the most recent years new warehouse tended to be built in census tracts containing a larger number of existing warehouses (what you call "clustering"). Does this cause overlap of NO₂ signal between warehouses located nearby? How do you deal with those situations? In the paper by Fioletov et al., where this wind-rotation method was introduced (to me at least), they mention that in case of sources located nearby there is possibility of multiple-count (in that case this referred to emission estimation). How do you see this in your case? Are the clustered areas somewhat overcounted? Or is this not an issue for some reason? Maybe, it would be useful to see a map or some analysis for one case where you have multiple warehouses close to each other, and how the rotation and calculation of near-warehouse enhancement work out.
- Figure 3A/S4. Maybe it could be useful to show the correlation coefficient of these linear regressions as well, in addition to the slope.
- Figure S6. It looks like the slope decreases with decreasing the buffer area radius. Is there a reason for that?
- P8 "...the proportion of ethnoracial minorities in tracts with warehouse(s) is smaller than the state median in a few states (e.g., Idaho, Montana, West Virginia; Figure S8)". Is this perhaps because of these states have a very large majority of white population?
- P8 (second paragraph). "Despite Blacks or African Americans living in close proximity to a greater

number of warehouses and associated loading docks and parking spaces (Figures 4A, S7), on a national scale, this population group faces lower NO₂ than average near-warehouse levels (-2.1%).": Could this be explained also by the fact the NO₂ levels away from warehouse are relatively high for Blacks or African Americans, maybe in large cities. I am not familiar with the general distribution of difference ethno-racial groups in US, but you might be able to confirm/deny that.

- As an overall comment: the second part of this manuscript somewhat overlaps with a previous paper by Demetillo et al. (2021) which deals with air pollution equalities in US cities. It would be beneficial to put the results presented here in the context of this previous work (maybe in the discussion). How much air pollution inequality is near-warehouse related and how much generally driven by the vicinity of traffic emissions in general or other factors?
- P10 last sentence: It would be useful I think to add an example of cluster (as I mention before) already in this paper to make the conclusion more convincing.
- Technical comments: some links from the reference list link direct to a Zotero page, maybe the proper paper link could be added.

Next time please add line numbers to the manuscript for an easier review.

Overall, the paper is scientifically sound and relevant. On the other hand there are plenty of excellent papers that do not make the cut on such high level journal. I leave to the editor to decide whether the results are striking enough to be published on Nat Comm.

Responses to reviewers

Reviewer #1 (Remarks to the Author):

Summary

This exploratory study assesses correlations between cross-sectional satellite measured NO₂ concentrations, traffic and warehousing across the United States, and whether warehouse locations are distributed unequally among different demographic groups. Using straightforward spatial statistics, the authors find that warehouses are correlated with elevated NO₂ concentrations and disproportionate exposures on non-white populations. I have no major comments for the manuscript. The authors clearly describe the major limitations of their analyses and provide reasonable justifications for their findings. I provide some minor comments to improve clarity.

We thank the reviewer for these helpful and constructive comments. We have revised our manuscript to address their comments. In particular, we added text throughout the manuscript that was responsive to the comments and also included an additional supplementary figure to explore the impact that clustered warehouses have on our key results. Point-by-point responses are included below in blue and line number correspond to the numbers in the track-changed manuscript.

Introduction

- Please provide a citation for the statement made on lines 32-33 “warehousing and goods movement...fine particulate matter.”

We thank the reviewer for this suggestion. Although the literature on warehouse-sourced pollution is sparse, we have included a couple related references in the revised manuscript in Lines 38-39. Namely, Shearston et al. (2020), who examined air pollution impacts from the opening of a large delivery service warehouse in the South Bronx, and Lovas et al. (2021), who surveyed warehouse workers at logistics companies on occupational risks including those from diesel vehicles.

Main

- It would be helpful to explain why NO₂ concentrations increase almost exponentially between the 8th and 10th deciles for clustering in low population density warehouse locations.

The reviewer points out an interesting question regarding the large difference in TROPOMI-measured NO₂ between warehouses in the 8th-10th deciles in low population density environments. We agree that the change is noteworthy, and we further explored what might drive this difference based on other variables or factors considered in our analysis (see following figure).

Since the number of parking spaces and population density decrease between the 8th and 10th deciles, it is unlikely that the nearly exponential change of NO₂ comes from these warehouses having more passenger vehicle traffic or being located in areas with more people and therefore likely more NO_x emissions. Although the number of loading docks increases from the 8th to 9th decile, there is no increase between the number of loading docks between the 9th and 10th deciles, so it is unlikely that changes in loading docks would explain the increase of NO₂. Truck

traffic does increase non-linearly between the 8th and 10th deciles, so it is possible that these quasi-exponential increases in truck traffic associated with clustering could lead to the change in NO₂ pointed out by the reviewer. However, since this analysis is more speculative and the exponential change between these deciles in low population density environments is a relatively minor component of Figure 2B's analysis we chose not to include this hypothesis in our revised manuscript.

Interpretation of the figure on the left follows the low population density part of Figure 2B in the main text. Specifically, for warehouses partitioned into deciles based on clustering, we show the median and 95% confidence intervals are shown for (top left) TROPOMI NO₂, (top right) population density, (middle left) the number of loading docks at warehouses, (middle right) the number of parking spaces at warehouses, and (bottom left) truck VKT.

- Please explain the type of parking spaces that are available at the warehouses. Are they for trucks or for employees?

We thank the reviewer for drawing our attention to a point where we would have been more clear. Within the CoStar database, “parking spaces” correspond to stalls that are generally intended as parking for light-duty vehicles, while loading docks are level with the trailer of a semi-truck that allow for loading/unloading of the truck’s inventory (<https://www.costar.com/about/costar-glossary>). In the revised manuscript, we have added text to further clarify this distinction (Lines 283-287).

Methods

- Explain why they chose a 7km buffer to examine warehouse-related NO₂

Since NO_x emissions are primarily emitted as NO and thereafter converted to NO₂ (Lines 94-97), we did not expect to see the highest NO₂ levels at the warehouse. With this in mind, we evaluated near-warehouse NO₂ composites for a variety of buffers, eventually settling on ±7 km based on visual inspection after finding that this buffer captures the enhanced NO₂ downwind of warehouses. Note that we have also tested the impact of other sized buffers on results (Figure S3) and still found increased NO₂ on the basis of warehouse characteristics. In the revised manuscript, we have clarified that our choice of the ±7 km buffer was arbitrary but that we have tested the robustness of these results to other buffers (Lines 305-310). We also note that at the behest of another reviewer, we have changed our analysis of near-warehouse NO₂ to extend ±7 km radially outward (rather than in each cardinal direction) in the revised manuscript.

- With clustering of warehouses, how did the authors control for buffers for multiple warehouses located within the same buffer? Please explain.

In our original submission, we presented results in the aggregate (i.e., not accounting for multiple warehouses located within the same TROPOMI grid cell or buffer) such as in Figures 2A, S2, S8-S9 but also presented results controlling for different levels of clustering (e.g., Figures S2, 4A). When two or more warehouses occupy the same TROPOMI grid cell or buffer, it is not possible to disentangle the satellite-observed NO₂ attributable to different coincident warehouses given the tools used in this study, so we believe it is justified to present some results not explicitly controlling for clustered warehouses.

In the revised manuscript, we have clarified this approach in the methods (Lines 326-332). Additionally, since a similar question was raised by another reviewer and could also be raised by readers, we added a supplementary figure (Figure S3) that follows Figure 2A. This new analysis shows that even single warehouses (i.e. first decile in Figure S2) are associated with enhanced NO₂, although the magnitude of the enhancement is about ~5% less than the 17.9% enhancement found when examining all warehouses and quoted throughout the manuscript (Lines 104-111).

Discussion:

- Please provide additional data needs and knowledge gaps that future research can fill to answer questions about the association between warehousing and air pollution, as well as environmental justice implications of warehousing.

We thank the reviewer for this thoughtful suggestion. In the original submission, we mentioned several ways that future modeling efforts can be improved to more explicitly account for warehousing-related emissions (Lines 221-224).

In the revised manuscript we have added some additional ways that future research can help us better understand the impacts of warehousing, discussing the recently-launched TEMPO

instrument and targeted field campaigns of surface-level monitors that would complement satellite observations (Lines 234-240).

- Please describe whether or how warehousing activities changed during the COVID-19 (i.e., the time period for this study) so that readers can understand the generalizability of the study results to present day.

We thank the reviewer for this excellent comment. Because NO₂ changed during the pandemic due to factors beyond e-commerce and warehousing (e.g., shifts in commuting with light-duty vehicles, reduced or grounded air travel, etc.), it was not possible to isolate how changing warehousing activities alone impacted NO₂. Instead, we opted to examine NO₂ near warehouses in 2021 after urban activity had largely rebounded to pre-COVID levels.

However, limited literature suggests that e-commerce increased during the pandemic (e.g., Luo et al., 2023). We did not find any studies that indicated whether these trends had lingering effects in 2021 and beyond.

In the revised manuscript, we have mentioned the growth of e-commerce during the pandemic and stressed that our values represent 2021 values (Lines 257-260).

Methods:

- Please describe how or whether CoStar data undergo any QA/QC, whether this dataset is comprehensive and any other potential limitations of the dataset that might bias the results. CoStar does not publish quality control and assurance metrics related to their commercial real estate data, but we note that this dataset (1) is the most extensive commercial real estate database in the U.S. and (2) undergoes vetting and updates by their in-house researchers. We know of no other databases (free or proprietary) that provide nationally-consistent data on warehousing, so it is difficult to understand how differences in warehouse data would impact our results. Since CoStar is essentially an observational dataset created with the use of public records and drone, aerial, and field observations (Lines 276-278), even if another dataset existed, it would not be possible to determine whether CoStar or this other dataset are more accurate.

In the revised manuscript, we have further described the methods used to build the CoStar database, highlighted that the use of this dataset has precedent in the literature, and discussed the lack of other benchmark datasets to which we can compare CoStar (Lines 291-296).

- Please describe any limitations in the methods used to approximate NO₂ concentrations by combining them with wind data.

Since the wind data from ERA5 is assimilated from a number of datasets, it is prone to biases due to factors such as the dataset's spatial resolution or complex terrain (e.g., land-sea breezes) that we did not discuss in the original manuscript, and we thank the reviewer for drawing our attention to potential limitations regarding our wind-adjusted NO₂.

An global evaluation of the accuracy of ERA5's hourly wind estimates at 100-m relative to estimates from point in-situ observations revealed an average mean absolute error of 9 to 13% (Pryor & Barthelmie, 2021). Our daily wind estimates are averaged over a five-hour wind (Lines

316-317). If we assume that some of the average hourly errors reported by Pyror & Barthelmie are stochastic, averaging over several hours would likely reduce these errors.

In the revised manuscript we have mentioned the excellent performance of ERA5 winds against *in-situ* observations (Lines 314-316).

Supplementary File

- Please provide a full description of each figure so that they can be standalone figures. E.g., change “Same as Figure 1A in the main text...” to “The number of warehouses per per country in U.S. counties as of 2021

We have updated several figure captions in the supplement such that they can be interpreted without the captions shown in the main text.

- Figure s4: please add the decile number below each bin

We thank the reviewer for this helpful comment that will improve the interpretability of this supplementary figure. We have edited this figure to include the decile bin and reworked the caption in the revised manuscript.

Reviewer #2 (Remarks to the Author):

General Comment:

This paper examines the air pollution (using NO₂ as a proxy) and truck traffic impacts of warehouses across the United States. The authors report that warehouses enhance local NO₂ levels by ~20% on average and that these impacts are disproportionately shared by marginalized communities.

This is a very important and timely paper. However, (1) several methodological approaches are not clearly described and, more importantly (2) for the implications of this paper and the strong conclusions that the authors draw, I believe more rigorous analyses are needed.

We thank the reviewer for their careful review of our manuscript. In our revision, we have clarified our methods in several places throughout the text. We also included three new supplementary figures (Figure S3, S11, and S12) that add further credibility to assumptions made in our analysis and the issue of confounding the review points out. We have provided point-by-point responses to all the reviewer's comments and believe that these revisions have further improved our study's suitability for *Nature Communications*. Line numbers correspond to the numbers in the track-changed manuscript.

Below please find my specific comments.

1. Abstract “[...] an average near-warehouse NO₂ enhancement of nearly 20% and are disproportionately located in communities with above-average shares of Black, [...]”: According to Figure 4B Black residents are exposed to lower NO₂ on average.

The reviewer is correct that Black residents are exposed to lower near-warehouse NO₂ on average compared with overall near-warehouse NO₂; however, census tracts containing a warehouse or warehouses have 69.5% more Black or African American residents (Lines 164-167, Figure 4A), and there is an unmistakable increase in the proportion of the Black or African American population as the number of warehouses in each census tract increases (Figure 4A). For these reasons, we retained the original text in the abstract but also note that we provide a more nuanced discussion of the disproportionate presence of warehouses in Black and African American communities but reduced NO₂ (Lines 164ff, 181ff).

2. Abstract “Satellite data highlight the need for [...]”: This is slightly awkwardly phrased. I understand that the authors here would like to highlight the advantage of using satellite data, but the way this is phrased it takes away something from the importance of the need that the findings highlight. Maybe rephrase?

We thank the reviewer for this suggestion. We have reworked this sentence to highlight the need for regulation around this growing industry while also mentioning satellite data (Lines 25-26).

3. Why/how was 2010 selected as the baseline year?

We initially selected 2010 based on visual inspection of Figure 1B-F, noting that this baseline year roughly corresponds to when the economy began to grow again following the 2007-2008 financial crisis and after which we saw an uptick in new warehouse construction (Figure 1B). We acknowledge that this choice is somewhat subjective, but in the revised manuscript we have added some additional text and also calculated the significance of trends shown in Figure 1C-F (see our response to the reviewer's following comment).

4. Figures 1C-F: Other than just visually, were these changes somehow statistically examined? In our initial manuscript we only relied on visual inspection, but we thank the reviewer for this comment and motivating us to be more quantitative in our assessment of the trends. In the revised manuscript, we have quantified the significance of temporal trends of these time series using the Mann-Kendall test (Lines 413-415), finding that all trends in Figure 1C-F were significant (Lines 76-80).

5. When reading this, I first thought that the +/- 7km buffer meant a circle with a 7-km radius around the warehouse. However, looking at the graphs, I realize that this approach includes in the near-warehouse zone those grid cells a part of which (the center? Some proportion?) falls within the 7-km buffer, resulting in a square (and not a circle). Please clarify.

We apologize to the reviewer for the confusion over our use of the term "buffer." To clarify, our buffer is a square whose center is the warehouse and extends ± 7 grid cells in each cardinal direction (Lines 303-305). In the initial manuscript, we used the terms " ± 7 grid cells" and " ± 7 km" interchangeably. While the 7 grid cells are approximately 7 km, the exact distance seven grid cells represent is dependent on the latitude. Thus, in the revised manuscript, we changed the occurrences of "7 km" to "seven grid cells" (e.g., Lines 83-85, 144-148, 305, etc.) to increase clarity.

6. How did the authors treat impacts of warehouses with overlapping buffers? How often did that occur?

In our original submission, we both presented results in the aggregate (i.e., not accounting for multiple warehouses located within the same TROPOMI grid cell or buffer; e.g., Figure 2A, S2, S8-S9) but also presented results controlling for different levels of clustering (e.g., Figure S2, Figure 4A). Of the 149,075 warehouses in the U.S. included in this study, only 10,548 (~7%) are the sole warehouse in a given census tract, reflecting that warehouses are predominantly located in close proximity to other warehouses. However, when two or more warehouses occupy the same TROPOMI grid cell or buffer, it is not possible with the tools used in this study to disentangle the satellite-observed NO₂ attributable to different coincident warehouses, so we believe it justified to present some results not explicitly controlling for clustered warehouses.

In the revised manuscript, we clarified this approach in the methods (Line 326-328). Additionally, since this question was raised by another reviewer and could also be raised by readers, we additionally added a supplementary figure (Figure S3) that follows Figure 2A. This added analysis shows that even single warehouses (i.e. first decile in Figure S2) are associated with enhanced NO₂, although the magnitude of the enhancement is about ~5% less than the 17.9%

enhancement found when examining all warehouses and quoted throughout the manuscript (Lines 104-111).

7. How large was the area of the “upwind orthogonal edge of the composite”? Just 14 by 1 km²?

Yes, the reviewer is correct this upwind orthogonal is comprised of fourteen 1 x 1 km² grid cells to represent air entering the composite or system. We have clarified this in the revised manuscript and further described our rationale behind this definition (Lines 89-90).

8. “In our case, the maximum levels were displaced ~4 km downwind of warehouses.”: (a) On average? (b) What does “in our case” mean here?

We thank the reviewer for catching an instance where we could have been clearer. “In our case” referred to our specific analysis (i.e., active warehouses through 2021, 2021 TROPOMI data, etc.), but we acknowledge that it might present confusion for readers. We have removed this clause and also added “on average” to make it clear that this finding is from the composite analysis (Lines 85-89).

9. Page 4 towards the end of the page (2-4 lines from the end): But wouldn’t 4 km be too long for this process? NO₂ decay occurs at much shorter distances than that (as the authors also note in the discussion).

The plume spread from a NO_x source, such as a warehouse, depends on the prevailing winds and lifetime of NO₂ (which, in turn, depends on the sun angle, chemical regime, and meteorology). While we did report that distance-decay gradients for NO₂ can be as short as 100s of meters (Demetillo et al. 2021, which derives their insight from Karner et al., 2010¹ and Choi et al., 2012²), others have shown that satellite-measured NO₂ can peak as far as tens of km downwind of sources (e.g., Pommier, 2023; Goldberg et al. 2019). Our finding that NO₂ peaks around ~4 km downwind of warehouses is thus in line with these studies. We also note that warehousing-related traffic might not exclusively occur at the warehouses themselves but also on local, collector, and arterial roads near the warehouse; thus, the NO₂ plume would be subject to continual and incremental enhancement as it flows across the near-warehouse area.

In the revised manuscript, we have referenced the studies of Pommier and Goldberg et al. to illustrate that there is a range in where we expect peak satellite-measured NO₂ levels (Lines 98-100).

10. Figure 2: Why not transform NO₂ levels to ppm and report those? Wouldn’t ppm be more relevant towards informing policy since standards are set using ppm?

We agree with the reviewer that reporting near-warehouse NO₂ in parts per billion by volume would help inform policy-related discussions (e.g., related to attainment of National Ambient Air Quality Standards or NAAQS) as well as enable us to estimate health impact via a health impact assessment.

¹ <https://pubs.acs.org/doi/10.1021/es100008x>

² <https://doi.org/10.1016/j.atmosenv.2012.07.084>

The TROPOMI satellite instrument, however, provides us with tropospheric column-integrated NO₂ levels (Lines 299-301), not surface level concentrations. There are ways to combine satellite data with chemical transport models (CTM) to estimate satellite-derived surface-level concentrations (e.g., Cooper et al., 2022³); however, this type of estimate could introduce biases into our analysis based on the assumptions from the CTM related to meteorology, the location and intensity of emissions sources, and chemistry. Moreover, NO₂ estimated by a CTM would not be entirely independent from truck traffic, since truck emissions are input to the CTM.

In our original submission we discussed the utility of surface-level NO₂ associated with warehousing (Lines 221-224), and in the revised manuscript we have expanded upon this to describe that satellite data, to date, represent the most empirical way to surveil ambient NO₂ pollution with complete spatial coverage (Lines 230-232).

11. Figure 2B: From what analyses are these p-values extracted? Was that a regression of e.g., NO₂ ~ loading docks in the lowest population density tertile?

Yes, the reviewer is correct that the p-values and correlation coefficients were calculated from a regression of NO₂ and loading docks, clustering, or parking spaces for each of the three population density categories (Lines 119-121). We note that although we visualized the results in several figures, including Figure 2B, in deciles for ease of interpretation, these statistics were calculated not just with 10 points (i.e., one for each decile) but using nearly 50,000 points (i.e., ~150,000 warehouses discretized into three population density groups). In the revised manuscript we reiterated this point in Lines 425-428.

12. The sensitivity analyses restricting near-warehouse zones to smaller buffers is not described in the Methods. Please add.

We thank the reviewer for pointing out this inadvertent omission in our original submission. In the revised manuscript we included a describe of this sensitivity analysis in the Methods (Lines 307-310).

13. Figure 3A assumes linear associations. Were they linear indeed?

The reviewer raises an interesting question. We assumed that the relationship between truck traffic and warehouse characteristics shown in Figure 3A was linear but we did not formally quantify whether this assumption was correct in the initial submission.

In the revised manuscript, we now include a supplementary figure (Figure S12) and corresponding discussion (Lines 248-253) that are based on an analysis of the residuals of the regressions shown in Figure 3A. In summary, the residual errors generally follow a normal distribution. There are a couple decile VKT-warehouse characteristic instances where the distribution is slightly skewed (e.g., panel (C) for clustering, specifically the 10th decile), but generally this is not the case.

This evidence-backed assumption of a linearity is also advantageous for interpretation of our results: readers can easily understand the implications of a unit change in clustering, loading

³ <https://doi.org/10.1038/s41586-021-04229-0>

docks, or parking spaces on truck VKT whereas using more complex linear or non-linear models would not lend themselves to this straightforward interpretation.

14. Last sentence of page 8: So six states make all the difference for warehouse-related NO₂ levels among Black residents? Were these the same states in which the 20% of the warehouses is located?

We thank the reviewer for this thoughtful comment that connects a couple different results from our study. First, the six states where the Black or African American residents face lower near-warehouse NO₂ are VT, WV, SC, MS, FL, and DC. Twenty percent of warehouses are contained with just CA and TX (20184 and 12674 warehouses, respectively), so there is not overlap.

However, if we remove these six states and recalculate the relative difference in population-weighted NO₂ averaged near warehouses for different population subgroups and the overall population-weighted NO₂ near warehouses (i.e., Figure 4B), we indeed see that the Black or African American population's relative difference switches from negative to positive. We note however, that the relative difference, although positive without the inclusion of these six states, is still quite small (~1%; see following figure), especially compared to the relative difference for other population subgroups. This result suggests that there are other factors at play such as traffic volumes at warehouses located in predominantly Black or African American neighborhoods.

In the revised manuscript, we have commented on this result (Lines 192-195).

15. First sentence in the Discussion: This is just confounding. Could the authors repeat analyses adjusting for factors they think would confound these associations to obtain less biased estimates?

We thank the reviewer for pointing out the issue of confounding, which we agree could alter the near-warehouse NO₂ signal. In the revised manuscript, we leveraged the Neighborhood

Emission Mapping Operation (NEMO; Ma & Tong, 2022) to understand enhanced NO₂ near warehouses in areas with exogenous emitting activities. As we now describe in the text (Lines 335ff) NEMO provides total and sector-specific NO_x emissions at high spatial resolution (1 km x 1 km) and we specifically examined the near-warehouse NO₂ signal for warehouses with high versus low emission intensities from all sectors and specific sectors.

If the near-warehouse NO₂ signal was caused by exogenous factors, we might expect to *not* find a signal when we stratify and composite warehouses based on underlying emitting activity. Instead, we still find a signal regardless of the levels of underlying emitting activity (Lines 196ff, Figure S11). In this addition, we note that the exact magnitude of the NO₂ enhancement varies based on underlying emission intensity and the dominant emitting sector, but the enhancements are largely within $\pm 10\%$ of the 17.9% enhancement quoted elsewhere in the text.

16. If I understood correctly, only areas with warehouses were included in analyses, yes? And the only comparison was with that upwind orthogonal edge? Why instead not match locations with warehouses to locations without warehouses but similar characteristics otherwise (e.g., population density, land use, building density, km of roads/highways, etc) to compare NO₂ levels? Are the upwind orthogonal edges exchangeable to the buffers in terms of potential confounders? If there are any systematic differences, then the reported results could be substantially biased.

Yes, the reviewer is correct that we examined NO₂, traffic, and demographics near warehouses (using a series of different definitions for what constitutes “near warehouse”) and defined the enhancement relative to the upwind orthogonal edge. We stress that the composites have been adjusted for the prevailing wind direction, which we expect would minimize the warehousing contribution on the composite.

The reviewer’s helpful comment is related to the issue of confounding. There are several potential ways to address confounding, and we tried to construct our analysis in a way that minimizes this influence. Still, we acknowledge that we could do more to explore how confounding might impact results. One way to do so, as the reviewer suggests, might be to match near-warehouse NO₂ with NO₂ located in similar environments (e.g., traffic, density, etc.) without warehouses. This pairing, though, might introduce biases to our analysis as there is no perfect proxy for NO₂. For example, while population density is correlated with NO₂, oil and gas extraction and associated emissions can take place in areas with low population density and traffic.

Another way to address the issue of confounding is by examining the near-warehouse NO₂ enhancement for warehouses located in different environments in terms of the intensity of emissions and dominant sectors. If the near-warehouse NO₂ signal we observed was due to confounding factors from colocated emission sources, we might expect to find no signal when emissions (or emissions from certain sectors) are low. We tested this hypothesis and the reviewer’s overall question about potential confounders using a high-resolution NO_x emissions inventory (Section 4.3). As we discussed in our response to the reviewer’s previous question, we still found a signal regardless of the levels of underlying emitting activity, although the exact

magnitude of the NO₂ enhancement varies based on underlying emission intensity and the dominant emitting sector (Lines 196ff, Figure S11).

17. There are many warehouses missing data on loading docks and number of parking spots. How were these warehouses treated in the analysis? Were they excluded? And was missingness informative based on some key characteristics (e.g., population density or land use)? If so, this could also substantially bias results.

We thank the reviewer for pointing out the potential for missing information on loading docks and parking spots to bias our results. We agree with this possibility, and in the revision we explored whether warehouses with missing data on loading docks and parking spaces were correlated with TROPOMI NO₂, population density, traffic, and demographics—all key variables used in our study (note that since other land-use terms were not a focus in this study we did not consider how they might correspond with the missing warehouse data).

In summary, we found the correlation between whether warehouses have missing loading dock or parking space information was generally $-0.1 \leq r_s \leq 0.1$. The one exception was $r_s(\text{missing parking spaces, truck VKT}) = -0.11$.

Given that <10% of the variability of TROPOMI NO₂, demographics, population density, and truck traffic is explained by the presence of data on loading docks or parking spaces, it is unlikely that the missing data would impact our key results. We have added some discussion of this potential in the Discussion (Section 3) of the revised manuscript (Lines 241-248).

18. I believe the TROPOMI overpass time is sometime mid-day, is that true? If so, how could that impact the results estimated here? I would assume that warehouse-related traffic would peak overnight or early morning. Could then using TROPOMI potentially result in underestimation?

The reviewer is correct that TROPOMI has an overpass time during midday, specifically around 1330 hours local time. We included this information in our revised manuscript (Lines 299-301). Since heavy-duty vehicle traffic peaks during the mid-morning (Harley et al., 2005), it is therefore likely that TROPOMI underestimates near-warehouse NO₂. In the revised manuscript we discuss this hypothesis, and we thank the reviewer for giving us this idea (Lines 200-202, 234-238).

19. Section 4.3, last two sentences: I am sorry, but I cannot understand this analysis... Maybe include an equation to clarify? But also, how was significance evaluated in this analysis? Or was it not evaluated? This also has important implications for interpretation of results.

We regret the lack of clarity in this section. In the revised manuscript we have added an equation to describe this analysis and also additional text to complement this equation (Lines 377ff).

Since the analysis in Section 4.4 that is shown in Figure 3B is the relative difference between two population-weighted NO₂ levels, we are not aware of a method to calculate whether the difference between two singular values is significant. We have included this caveat in the revised text (Lines 429-432).

20. Limitations?

In our original manuscript, we included some limitations related to our methodological approach such as exogenous activity near warehouses contributing to the NO₂ signal (Lines 196ff) but agree that we could provide more limitations to appropriately caveat our methods and findings.

In the revised manuscript, we have included several other limitations such as the potential for TROPOMI to underestimate near-warehouse NO₂ on account of its overpass time (Lines 200-202), missing warehouse characteristics from CoStar (Lines 241-248), and the linear assumption of our truck traffic-warehouse analysis (Lines 248-253).

Reviewer #3 (Remarks to the Author):

Review of the manuscript “Satellite data uncover nationwide air pollution impacts from warehousing” by Kerr et al.

The manuscript describes an approach to relate satellite-based nitrogen dioxide observations (indicator of air pollution) near warehouses in the contiguous USA to warehouse-related parameters such as number of loading docks, parking spaces and clustering. In addition, the author analyze the demographic composition near the US warehouses to assess how the

near-warehouse NO₂ levels are distributed among different ethno-racial groups. The quality of the data used is appropriate for such analysis and the methods applied are proper and mostly support the conclusions. The results are of significance as warehouse-related activities are on the rise in recent years and warehouse-related activities and traffic-related air pollution might have substantial effects on local population, especially from certain ethno-racial groups in the country. There are nevertheless some weaknesses in my opinion in the way the data and results support the conclusions.

We are grateful to the reviewer for their careful and thoughtful reviews of our manuscript and suggestions for ways to strengthen our study. We agree with the reviewer that there were some weaknesses in the initial submission of this study. In the revised manuscript, we have conducted additional analyses to strengthen our findings. Specifically, we wish to point out that we now consider a high-resolution emissions inventory to help understand the NO₂ signal in areas emissions from other sources, such as the oil and gas industry which the reviewer points out.

Below we detail how we addressed all the reviewer's comments and amended our manuscript in ways that were responsive to the weaknesses pointed out by the reviewer. Line numbers correspond to the numbers in the track-changed manuscript.

Specific comments:

- Title and intro: I think that the fact that analysis focuses on the US should be specified in the title. "nationwide" could refer to any "nation" unless you have a very USA-centric view of the world. The same applies to the introduction (third paragraph) where the authors use the word "states" twice to refer to US states. Please specify this more clearly.

We thank the review for drawing our attention to these instances in the text. We agree that generically referring to "nation" and "states" is not clear, and we have changed these occurrences in the revised manuscript (e.g., Lines 1, 45-46, etc.).

- Sect. 2 first paragraph: You mention that warehouses in rural areas are often collocated with oil exploration and production facilities. Could then the NO₂-signal in such situations be mixed with possible emissions from oil extraction/processing activities? How do you make sure these signals are distinguishable? I suppose the authors assumed that the rotation procedure sufficiently isolates the contribution from different sources, but what happens when other sources are within or close to the near-warehouse definition of 7km? Could you provide for example a plot of the rotated and not-rotated NO₂-map for Bakken, and explain how/whether those signal overlap and how you treat such cases in your analysis?

We agree with the reviewer that emitting activities collocated with warehousing could alter the near-warehouse NO₂ signal. In the original submission, we assumed that generating composites of NO₂ around the nearly 150,000 warehouses would smooth out over some "noise" (i.e., exogenous emissions); however, in the revised manuscript we explored whether this assumption held for different levels of underlying emitting activity.

Instead of providing a map of the rotated versus unrotated NO₂ signal for the Bakken and the oil and gas operations in that region, we systematically examined how the NO₂ signal is altered by different intensities of underlying emissions for (1) total NO_x emissions and (2) sector-specific

emissions by leveraging the Neighborhood Emission Mapping Operation (NEMO; Ma & Tong, 2022). As we now describe in the text (Lines 335-347), NEMO provides total and sector-specific NO_x emissions at high spatial resolution (1 km x 1 km) and we specifically examined the near-warehouse NO₂ signal for warehouses with high versus low emission intensities from all sectors and specific sectors. If the near-warehouse NO₂ signal was caused by exogenous factors, we might expect to not find a signal when we stratify and composite warehouses based on underlying emitting activity. We still find a signal regardless of the levels of underlying emitting activity (Lines 196ff, Figure S11). We note that the exact magnitude of the NO₂ enhancement varies based on underlying emission intensity and the dominant emitting sector (Lines 196ff, Figure S11), but the enhancements are largely within $\pm 10\%$ of the 17.9% enhancement quoted elsewhere in the text.

- Related to the previous point, you mention that in the most recent years new warehouse tended to be built in census tracts containing a larger number of existing warehouses (what you call “clustering”). Does this cause overlap of NO₂ signal between warehouses located nearby? How do you deal with those situations? In the paper by Fioletov et al., where this wind-rotation method was introduced (to me at least), they mention that in case of sources located nearby there is possibility of multiple-count (in that case this referred to emission estimation). How do you see this in your case? Are the clustered areas somewhat overcounted? Or is this not an issue for some reason? Maybe, it would be useful to see a map or some analysis for one case where you have multiple warehouses close to each other, and how the rotation and calculation of near-warehouse enhancement work out.

We thank the reviewer for raising an interesting point regarding the implications of clustered warehouses on the NO₂ signal. When there were multiple warehouses within the same 1 km x 1 km TROPOMI grid cell or within the same ~15 km x 15 km near-warehouse area investigated in our study, each warehouse *still* counted towards aggregate statistics or composites. We provided additional clarification about our approach in the methods (Lines 326-332). This methodological choice stems from our inability to ascribe a certain portion of the satellite NO₂ signal to a particular warehouse or traffic related to that warehouse, and we mentioned ways that future studies might leverage spatial allocation of emissions in photochemical models to address this limitation (Lines 228-230).

Still, the reviewer’s point is valid. We ultimately decided not to provide an example for a particular cluster of warehouses since our analysis examines warehouses in the aggregate. Rather, we provided an analysis of the NO₂ signal at individual (non-clustered) warehouses versus at warehouses with various degrees of clustering (Figure S3) and corresponding discussion (Lines 104-111).

- Figure S6. It looks like the slope decreases with decreasing the buffer area radius. Is there a reason for that?

We thank the reviewer for noticing this result and drawing our attention to it. We agree that, for areas with similar baseline volumes of traffic, the slope between truck VKT and warehouse characteristics indeed decreases as we restrict our analysis to smaller buffers around warehouses. The most likely explanation is that as we restrict the number of grid cells that

constitute the near-warehouse area, we, by definition, reduce the total traffic and truck traffic. This is because truck traffic measured in truck kilometers traveled *summed over* the near-warehouse area. We have provided this explanation in the revised manuscript (Lines 158-160).

- P8 "...the proportion of ethnoracial minorities in tracts with warehouse(s) is smaller than the state median in a few states (e.g., Idaho, Montana, West Virginia; Figure S8)". Is this perhaps because of these states have a very large majority of white population?

The reviewer is correct that several of the states where we found conflicting results regarding which populations live in closer proximity to warehouses generally have smaller populations of ethnic-racial minorities and/or have predominantly sparse, rural populations. Both of these factors could contribute to the unexpected results for states such as Idaho and Montana in Figure S9.

Our study's goal is to provide the first nationwide study of warehousing, pollution, and equity rather than provide extensive results for individual states, and we hope future research following the example provided by De Souza et al. (2022)⁴, who focused on warehousing and environmental justice in California, might provide state-specific studies. We mention this excellent point made by the reviewer in the revised manuscript and invite future state-specific research (Lines 177-180).

- P8 (second paragraph). "Despite Blacks or African Americans living in close proximity to a greater number of warehouses and associated loading docks and parking spaces (Figures 4A, S7), on a national scale, this population group faces lower NO₂ than average near-warehouse levels (-2.1%)." : Could this be explained also by the fact the NO₂ levels away from warehouse are relatively high for Blacks or African Americans, maybe in large cities. I am not familiar with the general distribution of difference ethno-racial groups in US, but you might be able to confirm/deny that.

We appreciate the reviewer focusing our attention on this unexpected finding regarding lower NO₂ for the Black or African American population. The reviewer is correct that the Black or African American population faces higher NO₂ levels from sources other than warehouses (e.g., Kerr et al., 2021, Kerr et al., 2023). However, since the comparison in Figure 4B was from population-weighted NO₂ near warehouses for the Black or African American population subgroup *to the overall population-weighted NO₂ levels near warehouses*, this analysis would not account for the fact that this marginalized population faces high NO₂ levels away from warehouses.

In response to another reviewer comment, we removed states where Black or African American populations faced lower near-warehouse NO₂ (from Figure S10) and recalculated population-weighted NO₂ levels for different population subgroups. With this approach we found that the Black or African American population experienced slightly higher NO₂ than average near-warehouse levels (Lines 192-195).

⁴ <https://doi.org/10.1016/j.jtrangeo.2022.103440>

We believe that further investigation as to why the near-warehouse Black or African American population faces lower NO₂ on average than overall near-warehouse populations is beyond the scope of our study but we do offer a hypothesis on what might cause this result relating to the geographic distribution of NO₂ and the Black or African American population (Lines 214-218).

- As an overall comment: the second part of this manuscript somewhat overlaps with a previous paper by Demetillo et al. (2021) which deals with air pollution equalities in US cities. It would be beneficial to put the results presented here in the context of this previous work (maybe in the discussion). How much air pollution inequality is near-warehouse related and how much generally driven by the vicinity of traffic emissions in general or other factors?

We thank the reviewer for drawing our attention to Demetillo et al. (2021). We cited this paper along with another excellent study by the same lead author (Demetillo et al., 2020) throughout our study. We agree with the reviewer that our results build on those of Demetillo by showing how one specific industry/source sub-sector—that is, warehousing—is associated with inequities in NO₂ among different racial-ethnic populations.

In the Discussion of the revised manuscript we have added additional commentary contextualizing our study in light of what has previously been shown in Demetillo et al., 2020, 2021 and other previous studies (Lines 209ff).

- P10 last sentence: It would be useful I think to add an example of cluster (as I mention before) already in this paper to make the conclusion more convincing.

We thank the reviewer for this idea and refer them to their previous comment about warehouse clustering and our systematic evaluation of the near-warehouse NO₂ signal near warehouses based on density or clustering.

- Technical comments: some links from the reference list link direct to a Zotero page, maybe the proper paper link could be added.

We thank the reviewer for drawing our attention to this inadvertent mistake. We believe we have fixed the Zotero links in the resubmission and will work with production staff to ensure that links point to the appropriate places when our study is published.

Next time please add line numbers to the manuscript for an easier review.

Line numbers have been added in the revised manuscript.

REVIEWERS' COMMENTS

Reviewer #1 (Remarks to the Author):

The authors have sufficiently responded to the reviewer comments.

Reviewer #2 (Remarks to the Author):

Thank you for carefully addressing my comments!

Reviewer #3 (Remarks to the Author):

The authors addressed carefully all the comments and from my side the paper can be accepted as is.